# Investigation of the Fuzzy Complex between RSV Nucleoprotein and Phosphoprotein to Optimize an Inhibition Assay by Fluorescence Polarization

**DOI:** 10.3390/ijms24010569

**Published:** 2022-12-29

**Authors:** Silva Khodjoyan, Deborha Morissette, Fortune Hontonnou, Luis Checa Ruano, Charles-Adrien Richard, Olivier Sperandio, Jean-François Eléouët, Marie Galloux, Philippe Durand, Stéphanie Deville-Foillard, Christina Sizun

**Affiliations:** 1Institut de Chimie des Substances Naturelles, CNRS, Université Paris Saclay, F-91190 Gif-sur-Yvette, France; 2Virologie et Immunologie Moléculaires, INRAE, Université Paris-Saclay, F-78350 Jouy-en-Josas, France; 3Structural Bioinformatics Unit, Department of Structural Biology and Chemistry, Institut Pasteur, Université de Paris, CNRS UMR3528, F-75015 Paris, France; 4Collège Doctoral, Sorbonne Université, F-75005 Paris, France

**Keywords:** respiratory syncytial virus, phosphoprotein, nucleoprotein, replication complex, fuzzy complex, protein–protein interaction, PPI inhibition, nuclear magnetic resonance, fluorescence anisotropy, B-cyano BODIPYs

## Abstract

The interaction between Respiratory Syncytial Virus phosphoprotein P and nucleoprotein N is essential for the formation of the holo RSV polymerase that carries out replication. In vitro screening of antivirals targeting the N-P protein interaction requires a molecular interaction model, ideally consisting of a complex between N protein and a short peptide corresponding to the C-terminal tail of the P protein. However, the flexibility of C-terminal P peptides as well as their phosphorylation status play a role in binding and may bias the outcome of an inhibition assay. We therefore investigated binding affinities and dynamics of this interaction by testing two N protein constructs and P peptides of different lengths and composition, using nuclear magnetic resonance and fluorescence polarization (FP). We show that, although the last C-terminal Phe_241_ residue is the main determinant for anchoring P to N, only longer peptides afford sub-micromolar affinity, despite increasing mobility towards the N-terminus. We investigated competitive binding by peptides and small compounds, including molecules used as fluorescent labels in FP. Based on these results, we draw optimized parameters for a robust RSV N-P inhibition assay and validated this assay with the M76 molecule, which displays antiviral properties, for further screening of chemical libraries.

## 1. Introduction

Respiratory Syncytial virus (RSV) is the most common pathogen for acute pediatric low respiratory tract infections (ALRI) and bronchiolitis [1]. RSV is an important cause of death in infants in developing countries and a substantial burden on healthcare systems and hospitals worldwide. In 2005, RSV led to ~33 million RSV-associated ALRI episodes, 3 million hospitalizations and 55,000–199,000 deaths in children younger than 5 years [2,3]. RSV also represents a still underestimated risk of severe infection and mortality for immunocompromised and elderly persons [4].

There is still no licensed human RSV vaccine, even after more than 6 decades of research. The monoclonal antibody Palivizumab, which is not efficient against ongoing infection, has been administered for prophylaxis only to high-risk infants, due to high cost and limited efficacy [5,6,7]. Very recently Nirsevimab, a long-acting antibody for similar applications [8], was approved by the European Medicines Agency. More than 30 RSV prevention candidates are currently in clinical development [9,10]. The licensed therapeutic arsenal against RSV is even more limited. Ribavirin, a nucleoside analogue, has a low therapeutic index and is only used for high risk patients. Several drug molecules, antibodies and a siRNA, targeting the fusion process or the viral replication and transcription machinery, have recently entered clinical trials [5,7,11,12]. Fusion inhibitors targeting the RSV F protein, like the small compounds JNJ-53718678 [13], BTA-C585 (clinical trial NCT02718937), GS-5806 [14], and the ALX-0171 nanobody [15] completed phase 2 clinical trials. The viral RNA polymerase inhibitor ALS-008176 [16] also completed a phase 2 trial. Although promising, these drugs elicit known resistances, which calls for alternatives and/or combination therapies.

RSV is a nonsegmented single-stranded negative-sense RNA virus of the *Mononegavirales* order, *Pneumoviridae* family, and *Orthopneumovirus* genus [17]. Its genome consists of an RNA molecule enchased in a sheath made of RSV nucleoprotein (N protein), forming a helical ribonucleoprotein complex termed nucleocapsid [18,19,20]. Replication of the viral genome as well as transcription of viral mRNAs are carried out by the viral RNA polymerase machinery. The apo polymerase consists of the RSV L protein, a large multifunctional catalytic subunit, and its essential co-factor, the RSV phosphoprotein (P protein). The P protein mediates recognition of the genomic material by the polymerase via direct binding to the N protein [21,22,23]. This interaction between P and N proteins was also found to drive the formation of cytoplasmic condensates, which are involved in RSV replication [24,25]. Although the primary function of RSV P protein consists of tethering the polymerase onto its template, P is a multifunctional protein with several binding partners. These properties are closely linked to its mainly disordered structure, outside a short central tetramerization domain P_OD_ (Figure 1A) [26,27,28]. The N-terminal domain of P (P_NTD_) is nearly fully disordered and contains several short linear motifs that are recognized by different viral and cellular proteins [29]. Of note, the N-terminus of P is involved in the formation of the N^0^-P complex, where P serves as a chaperone for RNA-free N protein (N^0^) and maintains N competent for encapsidation of genomic RNA [30]. The C-terminal domain of P (P_CTD_) contains a large transiently structured region with α-helical propensity (P_Cα_) [27], which is the binding site of the RSV L protein [23]. The 35–40 amino acid long C-terminal tail (P_Ctail_) is fully disordered, and the 10 last amino acids were reported to be involved in the N-P interaction relevant for the holo polymerase complex [21,26,31].

The RSV N protein is an RNA-binding protein. When overexpressed as a recombinant protein in *E. coli,* N binds to bacterial RNA and forms ring-shaped N-RNA complexes comprising 10–11 N protomers (Figure 1B) [32]. The N protein consists of two globular N-and C-terminal domains, N_NTD_ and N_CTD_, connected by a hinge, where RNA binds (Figure 1B). These two domains are flanked by flexible N- and C-terminal arms. The N-terminal arm (35 residues) engages into inter-protomer interactions in N-RNA rings [32]. N_NTD_ contains the binding site of P_Ctail_ [21].

In contrast to full-length N, recombinant N_NTD_ is produced as a monomeric and RNA-free protein. We previously provided a high resolution structural basis for the N-P interaction with X-ray crystal structures of complexes between N_NTD_ and C-terminal P peptides, notably with the 2-mer Asp_240_Phe_241_ (P_2_ peptide in Figure 1A,C) [33]. The C-terminal Phe_241_ residue of P_2_ is anchored into a hydrophobic pocket at the surface of N (Figure 1C). Strikingly, for longer peptides, only this Phe_241_ residue was ordered in the crystal structures and adopted a well-defined position (Figure 1C). Nevertheless, there is evidence that the N-binding region of P extends beyond Phe_241_ in P_Ctail_. A minigenome assay showed that in vitro replication of RSV was attenuated for the single Leu_238_Ala and the double Glu_239_Ala/Asp_240_Ala P mutants [21]. This result is in line with in vitro binding assays performed with recombinant N protein and GST-fused P mutants, which showed that up to 9 C-terminal P residues are necessary for N-binding [31]. This was corroborated by affinity measurements carried out by NMR with two C-terminal P peptides, P_2_ and the 12-mer P_12_ (Figure 1A). P_12_ displayed a 100-fold lower dissociation constant K_d_ (54 ± 9 µM) than P_2_ (4.2 ± 1 mM) [33]. We hypothesized that P_Ctail_ could make multiple contacts with the surface of N, increasing avidity of P and contributing to affinity. Since the position of these contacts is not well defined, N_NTD_-P_CTD_ can be considered as a fuzzy complex [34,35]. Electrostatic interactions between the highly acidic P_Ctail_ and positively charged residues that are exposed at the surface of N around the P-binding pocket likely provide additional binding contribution. In cells, the negative charge of P_Ctail_ is increased by constitutive phosphorylation of the Ser_232_ and Ser_237_ residues [36,37,38,39]. Phosphorylation was shown to modulate RSV replication in vivo [39]. However, the Ser_232_ phosphorylation site was reported to be dispensable for viral RNA transcription and virus replication in vitro [36,38,39].

Since the N_NTD_-P_CTD_ interaction is essential for the formation of the viral holo RNA polymerase complex and hence for RSV replication, we proposed that the RSV N_NTD_-P_CTD_ protein-protein interaction (PPI) could be a new drug target for PPI inhibitors. We previously showed that P_CTD_ could be displaced in vitro from N_NTD_ by 1-benzyl pyrazole 3,5-dicarboxylate derivatives, which mimic the sidechain and C-terminal carboxylate of Phe_241_ (Figure 1C) [33]. After chemical modification to improve its membrane passage, the M76 molecule (Figure 1D) was able to inhibit replication of recombinant fluorescent RSV in cells. Results from other groups reinforce this hypothesis. Hesperitin, a flavanone that reduced intracellular RSV replication [40], was also shown to bind to the P-binding pocket of N_NTD_ and to compete with the 11-mer P_11_ peptide [41]. The EDP-938 benzodiazepine, which demonstrated antiviral activity, elicited mutations on N close to the P-binding pocket, suggesting that it might be an N-P inhibitor [42].

Hesperitin and M76, which compete with P for N-binding in vitro and display antiviral activity [33,40,41], were identified from a limited set of molecules and different screening approaches: hesperitin was identified from a flavonoid series using an antiviral assay, whereas M76 was obtained from a subset of the ZINC database selected in silico with a stringent filter based on chemical structure. High-throughput screening (HTS) of chemical libraries would enable to more widely explore the potential of the RSV N_NTD_-P_CTD_ PPI as an antiviral target. This requires a relevant model of the interaction combined to a robust HTS method. We have shown previously that the complex between N_NTD_ and short C-terminal P peptides, derived from P_Ctail_, affords a minimal complex mimicking the interaction between the P protein and the RSV nucleocapsid [33]. Fluorescence anisotropy (FA) [43], or equivalently fluorescence polarization (FP) (see Materials and Methods), are compatible with HTS [44]. FP has already been applied in the case of the RSV N-P interaction to probe inhibition by hesperitin [41] and by P_Ctail_-derived peptides [45]. These FP measurements were performed under different experimental conditions, notably with different N protein constructs, different P peptides and different fluorescent labels. In this context, our aim was to obtain a deeper insight into the experimental requirements of an FP assay specific of the RSV N-P interaction. This led us to analyze the fuzzy complex formed between the N protein and the flexible P_Ctail_.

## 2. Results

### 2.1. Phosphorylation of an RSV P_Ctail_-Derived Peptide Increases Affinity for RSV N_NTD_

We previously analyzed the RSV N_NTD_-P_12_ complex using Nuclear Magnetic Resonance (NMR) [33]. As the isoelectric point of N_NTD_ is 7.8, we used MES pH 6.5 buffer to ensure the stability of the protein. This pH is also suitable for amide-detected 2D NMR experiments. The buffer contained salt (250 mM NaCl) to further stabilize N_NTD_ at protein concentrations required for NMR (50–100 µM). Under these experimental conditions we determined a K_d_ of 54 µM.

Here, as P protein is constitutively phosphorylated on Ser_232_ and Ser_237_ residues in cells, we first investigated the impact of phosphorylation on the affinity of the N_NTD_-P_Ctail_ complex. We performed an NMR titration experiment by adding increasing amounts of pP_11_ peptide, where Ser_232_ and Ser_237_ residues were replaced by O-phosphoserines (pSer, Figure 1A), to a solution of ^15^N-labeled N_NTD_, and measured 2D ^1^H-^15^N HSQC spectra (Figure 2A). We worked in the same buffer as with P_12_. At pH 6.5, the pSer sidechains are expected to be mostly negatively charged [46]. Addition of pP_11_ induced perturbations in the HSQC spectrum of N_NTD_, in particular chemical shift perturbations (CSPs). Saturation of this effect was reached after addition of ~6 molar equivalents of peptide. At this peptide:protein ratio, the protein can be considered to be in a fully bound form. We measured combined ^1^H and ^15^N amide CSPs (Equation (1)) for N_NTD_ signals in a fast chemical exchange regime, i.e., signals that shifted in the spectrum without broadening at intermediate titration points (e.g., Ile_129_ in Figure 2A). Significant CSPs were located in and around the P-binding pocket, confirming that pP_11_ binds to the previously determined P-binding epitope on N_NTD_ (Figure 2B,C) [33]. We fitted the NMR titration curves of these residues with a single site binding model to determine residue-specific K_d_s (Equation (2)), and calculated a mean K_d_ of 17.0 ± 4.5 µM from 26 individual values. Compared to the unphosphorylated P_12_ peptide, this represents a ~3-fold gain in affinity, which confirms the role of negative charges in P_Ctail_ for N-binding. Since measurements were made in a buffer with high ionic strength, the gain remains rather modest due to charge screening.

Overall, signals of bound N_NTD_ were broader than those of unbound N_NTD_ (Figure 2A). This can be explained by the increase of molecular weight, when the complex is formed. Line broadening at saturation was more marked for residues belonging to the P-binding site, suggesting exchange phenomena as additional sources of broadening. They may reflect local mobility of P in the complex and intermediate binding states. Some N_NTD_ signals with large chemical shift differences between bound and unbound states (e.g., Y_135_ in Figure 2A) were broadened at intermediate titration points. Signals were recovered when saturation was reached, indicating that they are in an intermediate exchange regime. These residues were also located in or in the close vicinity of the P-binding pocket, and thus report on the same binding mode as residues in fast exchange. From chemical shift differences measured for these residues we determined an exchange rate k_ex_ of ~800 s^−1^ (from Equation (3)) between free and bound N_NTD_, which indicates a fast dynamic process. Since k_ex_ (k_off_ + k_on_ × [N_NTD_]) is a combination of association (k_on_) and dissociation (k_off_) rates, the relative contributions of dissociation and association could not be assessed by this method.

### 2.2. Fluorescence Polarization Reveals a Potential Secondary Binding Site on RSV N_NTD_ for Fluorescein-Labeled RSV P_11_ Peptide 

2D Protein-observed NMR is a powerful tool that gives detailed structural and thermodynamic insight, but requires large sample amounts and is time-consuming. Acquisition of a single titration point in Figure 2 took 15 min. It is therefore not adapted for medium to high-throughput assays. Fluorescence polarization (FP) is more time-effective. FP is routinely used in screening assays for binding inhibitors [43,44,47,48]. We thus tested if we could transpose the experimental NMR conditions to FP measurements. We first wondered if the N_NTD_ construct, used for NMR, but also previously for isothermal titration calorimetry and surface plasmon resonance (SPR) experiments [33], was adapted for FP. As a general rule, fluorescence polarization increases with the apparent size of a fluorescent molecule. According to Perrin’s equation (Equation (5)), the theoretical FP of fluorescein (fluorescence lifetime ~4 ns) bound to N_NTD_ (26.2 kDa) is ~400 mFP. This is close to 500 mFP, the theoretical maximal FP value, and indicates that the size of N_NTD_ is compatible with an FP assay.

We started by labeling the pP_11_ peptide with N-terminally attached BODIPY FL, as described by Shapiro at al. [45]. However, we observed that the BODIPY FL-labeled peptide (BF_2_pP_11_) was prone to degradation, either via peptide bond cleavage or BODIPY FL-defluorination, under slightly acidic conditions. We then produced F_11_, another labelled peptide P_11_, by changing the fluorescent label to fluorescein-5-thiourea (FTU) by coupling with fluorescein-5-isothiocyanate (FITC). F_11_ proved to be stable. It is similar to the 6-carboxyfluorescein P peptide used by Sa et al. for competition with hesperitin [41]. To directly compare FP and NMR data, we made measurements in MES pH 6.5 buffer. Brij-35 detergent (0.01% *w*/*v*) was added below its critical micellar concentration (0.11% *w*/*v*) to prevent spurious binding of the peptide to the reading plates. To determine the binding affinity of F_11_ peptide for N_NTD_, we measured FP for 200 nM F_11_ by varying the concentration of N_NTD_, which ranged from 0.1 to 120 µM (Figure 3A). We tested buffer with and without salt.

In the presence of salt, FP measurements were stable for incubation times up to 48 h. FP of free F_11_ was 30–40 mFP. We fitted the binding curves with a single site binding model (Equation (10)). This yielded a maximal FP difference between bound and free F_11_ peptide ΔFP_max_ = 210–230 mFP and a dissociation constant K_d_ = 90 ± 15 µM (Figure 3A). Saturation of the binding curve could not be obtained within the applied N_NTD_ concentration range, and this may induce an error on K_d_ determination. However, the K_d_ has the same order of magnitude as that determined by NMR for the P_12_ peptide under similar buffer conditions (K_d_ = 54 µM) [33]. The experimental FP_max_ value was lower than the theoretical value (~400 mFP), suggesting that the fluorescent label displays some mobility with respect to the complex. As the label is attached to the N-terminus of the peptide and hence to the opposite side of the anchoring Phe_241_ residue, this would be in line with a fuzzy N_NTD_-P_CTD_ complex, where only the C-terminal Phe_241_ residue is well bound, whereas the rest of the P peptide remains disordered and only loosely attached.

In buffer without salt and in the presence of N_NTD_, F_11_ FP stabilized only after 30 min. Binding curves were obtained, but the FP data did not reach a plateau at high N_NTD_ concentration. Instead, FP displayed a linear dependence with increasing N_NTD_:F_11_ molar ratios, suggesting a second binding mode with low affinity (Figure 3B). We fitted the curves with a combination of a single binding site for the first mode and a linear contribution for the second mode. This yielded ΔFP_max_ = 175 mFP and K_d_ = 0.6 ± 0.1 µM. ΔFP_max_ was of the same order of magnitude in buffers with and without salt, indicating that the dynamic range of FP was not affected by salt. This suggests that the dynamics of the fluorescent probe were similar in these two conditions. In contrast, the affinity was significantly higher in the absence of salt, by 2 orders of magnitude, confirming the importance of electrostatic interactions for the N_NTD_-P_11_ complex. 

We previously showed by SPR that the M76 molecule (Figure 1D) could compete with P for N_NTD_ binding in vitro [33]. We therefore tested competition between M76 and F_11_ by FP. Measurements were done in salt-free buffer to take advantage of the higher F_11_-N_NTD_ affinity. The concentration of N_NTD_ was set to 5 µM to benefit from a high range of FP values, comprised between FP_min_ and ~80% FP_max_ [48]. With increasing M76 concentration (up to 100 µM), F_11_ FP progressively decreased due to F_11_ displacement from N_NTD_ by M76 (Figure 3C). However, even at high M76 concentration, FP did not reach the FP_min_ value. Saturation occurred at a high FP of ~105 mFP, suggesting that M76 was not able to fully displace F_11_. This corroborates the hypothesis of a secondary binding site on N_NTD_, for which M76 does not compete. We fitted the data with a Hill equation (Equation (12)), assuming that the maximal inhibition of F_11_ binding to the canonical site by M76 was obtained at saturation. We obtained an IC_50_ of 28 ± 5 µM. This was converted into an inhibition constant K_i_ of 2.5 µM using the IC50-to-Ki server [49], assuming a competitive mechanism and following values: K_d_ = 0.6 µM for F_11_, [N_NTD_] = 5 µM, and [F_11_] = 0.2 µM.

### 2.3. The Complex betweeen Fluorescein-Labeled C-Terminal P Peptides and Full-Length N Protein Provides a Robust Model of the RSV N-P Interaction for FP Measurements 

As the FP data obtained with N_NTD_ in salt-free buffer suggest the existence of a secondary binding site, we wanted to test if this was still the case with full-length RSV N protein. Wild-type N is produced in the form of ring shaped N-RNA complexes by overexpression in *E. coli* [32]. These rings mimic one turn of the helical nucleocapsid, which is the natural partner of P_Ctail_. We therefore chose to test N-RNA as an alternative N form. N-RNA complexes contain 10–11 protomers and 7 ribonucleotides per N protomer [32]. Due to the high molecular weight of N-RNA rings (~0.5 MDa), a high fluorescence polarization is expected.

We carried out FP measurements using the fluorescent F_11_ peptide and N-RNA complex. As N-RNA is more stable at pH 8.0 than at pH 6.5, we worked in Tris buffer. No salt was added to promote high affinity. The FP binding data could be fitted with a single site binding model (Figure 4A). We obtained ΔFP_max_ = 125–130 mFP, which is less than determined with the N_NTD_ construct, but again confirms the relative mobility of the fluorescent label. We determined a K_d_ of 7.6 ± 0.9 µM. Surprisingly, the K_d_ value obtained with N-RNA was one order of magnitude higher than with N_NTD_ (0.6 µM), suggesting that binding to N_NTD_ may not fully recapitulate binding to N-RNA.

We next performed inhibition experiments for the F_11_-(N-RNA) complex by M76 (Figure 4B). FP values decreased with increasing competitor concentration. At saturation, FP nearly reached the value of free F_11_, showing that the F_11_ peptide could be nearly completely displaced. Since the same concentration range was used for N_NTD_ and N-RNA, poor solubility of M76 can be ruled out to explain the saturation observed before complete displacement of F_11_ in the presence N_NTD_ (Figure 3C). This reinforces the hypothesis of a second binding site on N_NTD_, outside the canonical P-binding site. We fitted an IC_50_ of 12.6 µM (Figure 4B), which converts into a K_i_ of 2.4 µM (using K_d_ = 7.6 µM for F_11_, [N] = 10 µM and [F_11_] = 0.2 µM). The inhibition constant is similar to that obtained with N_NTD_ in salt-free buffer, which suggests that M76 binds in a similar way to the P-binding site on N_NTD_ and N-RNA.

M76 was dissolved in organic solvent such as ethanol or DMSO. This may affect the stability of the protein. We thus run a control experiment to evaluate the effect of solvent on FP measurements. We used the FTU-P_7E_ peptide F_7E_ (Table 1). Addition of up to 10% ethanol did not significantly affect FP measurements (Figure 4C). A slight decrease of FP was observed with 10% DMSO, but overall the FP assay is robust with respect to solvent. In summary, the N-RNA form appears to be suitable for FP measurements, as only specific binding of fluorescent P peptide and efficient inhibition by the M76 molecule were observed.

### 2.4. Measurement of the Binding Affinity of RSV Fluorescein-P Peptides for RSV N-RNA and Influence of Peptide Length on Fluorescent Label Mobility 

Fluorescent labels can only be attached to the N-terminus of the P peptides, since the C-terminus of P directly anchors onto the N protein via the Phe_241_ residue (Figure 1C). Since C-terminal P peptides are flexible [26], the fluorescent label displays relative mobility, as reflected by the moderate ΔFP_max_ values reported in the previous paragraphs. This raises the question about how the P peptide length impacts binding affinity and FP_max_. We thus designed a series of eight P peptides with increasing length, from 3 to 11 amino acids, and labeled them with FTU (Table 1). Moreover, to investigate the effect of phosphorylation, we produced phosphomimetic peptides, where either Ser_237_ alone or both Ser_232_ and Ser_237_ were replaced by glutamates.

We first measured binding of F_n_ peptides (200 nM) to N-RNA in Tris buffer by FP (Figure 5A,B). For F_n_ peptides with n ≥ 5, saturation of fluorescence anisotropy could be reached within the concentration range used for N-RNA, i.e., with 40 µM of total N protein concentration. The fully bound state of F_3_ and F_4_ peptides could not be obtained. We fitted the data with ΔFP_max_ and K_d_ as parameters. We observed that ΔFP_max_ decreased with the length of the peptide (Table 1), indicating that the dynamics of the fluorophore increase with its distance to the Phe_241_ anchor. Concomitantly, the affinity of F_n_ peptides increased with peptide length: K_d_s decreased from 62 µM (F_3_) to <1 µM (F_11SE_) (Table 1), highlighting the contribution of residues other than the two C-terminal P residues for N binding. In the 11-mer peptide series, phosphomimetic substitutions increased the affinity of F_11_, 9-fold for F_11SE_ and 20-fold for F_11EE_ (Figure 5B, Table 1). A closer inspection of the binding curves shows that F_11SE_ and F_11EE_ seem to display a second binding mode of weak affinity, which becomes visible once the canonical binding site has been saturated. For F_11SE_ this mode could be fitted with a linear contribution (Figure 5B, Table 1).

To validate fluorescent peptide binding to N-RNA, we carried out competition experiments with unlabeled P peptides. We used 10 µM of N protein. We first probed competition of the F_11_ peptide by three P peptides of different lengths: P_7E_, P_9E_ and P_11EE_ (Figure 5C). These phosphomimetic peptides were able to fully displace F_11_, with IC_50_ values ranging from 22 µM for the longest P_11EE_ peptide to 180 µM for the shortest P_7E_ peptide (Table 2). The trend of IC_50_ suggests that the efficiency of P peptides in competing with F_11_ increases with peptide length. This is in line with the increase of affinity observed with increasing length of fluorescent P peptides (Table 2). Inhibition constants, calculated with K_d_ values determined for F_11_, ranged from 6 to 75 µM (Table 2). The trend of K_i_ was the same as that of IC_50_. It is noteworthy that K_i_s of unlabeled peptides were higher than the K_d_s of their FITC-labeled counterparts, suggesting that the fluorescent peptides may display higher affinities than the unlabeled ones.

We next made competition experiments between three fluorescent and unlabeled peptides with identical amino acid sequences: F_5E_ vs. P_5E_, F_7E_ vs. P_7E_ and F_11_ vs. P_11_ (Figure 5D, Table 2). The unlabeled peptides were able to fully displace their fluorescent counterparts. IC_50_ and K_i_ values decreased with increasing peptide length, underlying the importance of P peptide length for inhibition of the N-P complex. Comparison of P_11_ with P_11EE_ for inhibition of F_11_ binding (Table 2) showed that P_11_ competed less efficiently than P_11EE_ with an IC_50_ of 86 versus 22 µM (K_i_ of 34 versus 6 µM). This confirms the role of additional negative charges for formation of the RSV N-P complex.

Finally, we probed inhibition by the M76 molecule of F_11_ and F_7E_ in complex with N-RNA. M76 IC_50_ and K_i_ values were in the µM range, but still lower than those of any tested unlabeled P peptide, indicating that M76 is a more potent inhibitor than the peptides, including P_11EE_. Moreover, M76 appears to be more potent to inhibit the N-F_11_ complex than the N-F_7E_ complex (K_i_ of 2.6 versus 8 µM). Although F_11_ and F_7E_ cannot be directly compared because of their different lengths, this finding suggests that M76 is less efficient at competing with a phosphomimetic peptide. A rationale for this would be that M76 directly competes with Phe_241_, but not with Ser/Glu_237_ that binds outside of the cavity.

It must be noted that measurements with N_NTD_ and N-RNA were done at different pH. Fluorescein fluorescence depends on pH, since the phenols and carboxylic acid groups can be ionized in a pH-dependent manner [50]. At pH 6.5, fluorescein dianion and monoanion forms are in a 1:1 equilibrium, whereas at pH 8.0 the dianion is predominant [51]. At pH 6.5 fluorescence emission at 490 nm is thus decreased by 30–40% as compared to pH 8 [52]. Nevertheless, FP should not be affected by pH, since fluorescence polarization is a ratio of emitted fluorescence intensities (Equation (4)). However, lower fluorescence leads to a lower signal-over-noise ratio at pH 6.5. In conclusion, N-RNA in a pH 8.0 buffer appears to be better suited for an FP inhibition assay than N_NTD_ in a pH 6.5 buffer.

### 2.5. Fluorescein Binds to the RSV P-Binding Site on RSV N Protein

Although unlabeled P peptides were able to displace their fluorescent counterparts from RSV N-RNA, we wondered if the difference between the K_i_ of F_n_ peptides and the K_d_ of unlabeled peptides could originate in N-RNA binding of the fluorescein label. We therefore measured fluorescein FP in the presence of N-RNA. We obtained a binding curve (Figure 6B). Complete saturation of fluorescein by N-RNA was not reached in the N concentration range used for the experiment. We tentatively extracted a K_d_ of 91 µM (Table 3) from these binding data, assuming that FP_max_ reaches the maximal theoretical value (500 mFP). The binding affinity was lower than that measured for F_n_ peptides, but still significant. The contribution of fluorescein binding to FP could be non-negligible, especially for the shorter F_3_ and F_4_ peptides, for which we determined K_d_s of 62 and 35 µM, respectively (Table 1). However, the affinity of fluorescein was lower by at least of one order of magnitude as compared to peptides longer than F_5E_. Hence, for these peptides, the fluorescent label is not expected to strongly compete with the peptide part. To test if fluorescein binds to the P-binding site on the N-RNA, we carried out a competition experiment with M76 (Figure 6C). M76 was able to displace fluorescein rather efficiently, with an IC_50_ of 27 µM and a K_i_ of 13 µM (Table 3), indicating that fluorescein indeed targets the P-binding site of the N protein.

To validate binding of fluorescein to the P-binding site, we made an NMR titration experiment with ^15^N-labeled N_NTD_, using experimental conditions similar to those of the titration of N_NTD_ by the pP_11_ peptide. Fluorescein induced CSPs located in the same region as those induced by pP_11_ (Figure 7A), indicating that fluorescein targets the P-binding site of N_NTD_ (Figure 7B,C). To get atomic details about the N_NTD_-fluorescein complex, we run docking experiments using 2 docking softwares: MOE with GBVI/WSA dG scoring function [53] and SMINA with Vinardo scoring function [54]. Two different poses of fluorescein inserted into the P-binding pocket were predicted (Figure 7D). Fluorescein was modeled in its open, fluorescent form. In all cases, one of the xanthene rings was inserted into the hydrophobic P-binding pocket of N_NTD_, and the benzoic acid was positioned similarly to the pyrazole ring of M76, forming salt bridges with R150 or R132 (Figure 7D,E).

To avoid interference of the fluorophore with FP measurements, we sought to lower the affinity of fluorescein for RSV N protein, while retaining fluorescence property, by chemical modifications. We produced methyl and ethyl esters of the benzoic acid to change the charge and the size of the molecule (Figure 6A). A series of FP measurements was made with different N-RNA concentrations. The two fluorescein derivatives also bound to N-RNA, but displayed 2-fold lower affinity (Figure 6B, Table 3). M76 was able to displace these two variants (Figure 6C). K_i_s were of the same order of magnitude at the K_i_ of fluorescein (Table 3). These binding and inhibition experiments suggest that the behavior of the two fluorescein derivatives is close that of the original molecule. Indeed, fluorescein methyl ester adopted docking poses similar to those of fluorescein. 

### 2.6. An FP Assay for the RSV N-P Interaction Using Full-Length N and BODIPY FL-Labeled P Peptides

The undesirable binding of fluorescein to N-RNA led us to reconsider the BODIPY FL as label for peptides, despite the preliminary mixed results obtained with this fluorophore. To get rid of the instability problems encountered with 4,4′ difluoro BODIPY FL scaffold, we considered its 4,4′ dicyano analogue reported as being more stable and with better photo-physical properties. Such derivatives were recently described as tags for bioconjugation [55,56], but, to our knowledge, not for an FP assay. While difluoro-**4** and dicyano BODIPY FL **5** also bound to N-RNA, FP values were markedly lower than those of fluorescein at the same N-RNA concentrations (Figure 6B), suggesting that BODIPY FL is less prone to bind. The K_d_ for **4**, obtained by fitting the ΔFP data with FP_max_ = 500 mFP, was also higher than that of fluorescein (Table 3). Altogether, these results show that the nature of the fluorophore must be taken into account in an FP assay with RSV N and P proteins, and that BODIPY FL is less prone than fluorescein to affect FP measurements. Moreover, as compared to difluoro BODIPY FL-labeled peptides, dicyano BODIPY FL-labeled peptides were chemically stable (either under the acidic conditions used for their purification by HPLC or under the conditions of the FP assay). To assess the properties of BODIPY FL as a fluorescent label for P peptides, we produced a new series of eight dicyano BODIPY FL-labeled (BCN_n_) peptides of variable length (Table 4) and tested them for N-RNA binding.

We first performed FP measurements in salt-free Tris pH 8 buffer (Figure 8A). We measured binding curves using 200 nM of fluorescent peptides. Measurements were stable within a time span of 1 h. Saturation by N-RNA was reached for all peptides, except for the shortest one BCN_5E_. Overall, the dynamic range of FP was more extended for BCN_n_ peptides than for F_n_ peptides, with ΔFP_max_ of 270–430 mFP, as compared to 95–315 mFP for F_n_ peptides (Table 4). Fitted K_d_s were similar to those of their equivalent F_n_ peptides, suggesting that the fluorophore does not contribute to the affinity of P peptides longer than 5-mers. Overall, affinities increased with the length of the peptides. The BCN_n_ peptide series exhibited a similar trend to that of the F_n_ series, highlighting the binding contribution of residues beyond Asp_240_ and Phe_241_. A gain of affinity was observed for BCN_11EE_ vs. BCN_11SE_, showing that the phosphomimetic at position 232 contributes to binding.

To evaluate the quality of FP measurements, we performed a Z’ assay [57,58] with the BCN_10EE_ peptide. An N-RNA concentration of 1 µM afforded a high dynamic range (ΔFP_max_ = 220 ± 13 mFP, with FP_min_ = 9 ± 3 mFP). We determined a Z’ value of 0.78 from seven data points, which validates the BCN_10EE_ probe. We further tested robustness with respect to solvent. FP values were not significantly affected by addition of up to 5% ethanol and 10% DMSO.

As compared to salt-free buffer, the ΔFP_max_ range in salty buffer (100 mM NaCl) was significantly reduced for the long BCN_10EE_, BCN_11SE_ and BCN_11EE_ peptides (Figure 8B, Table 4). This is similar to what was observed for the F_11_ peptide. The fluorescent label of long P peptides thus appears to be more mobile under salty conditions, likely due to screening of electrostatic interactions involving acidic amino acids of P peptides. This was not observed for the shorter B_6E_ and B_8E_ peptides. However, it must be noted that the binding curves of BCN_6E_ and BCN_8E_ did not reach a plateau within the N-RNA concentration range used in our FP experiments, which may affect the fitted parameters. The affinity of BCN_6E_ and BCN_8E_ peptides decreased 5-fold in salty buffer, and that of BCN_10EE_ 8-fold, as expected by screening of electrostatic interactions in a higher ionic strength buffer. Intriguingly, similar affinities were measured for BCN_11EE_ and BCN_11SE_ under both salt conditions, despite the phosphomimetic substitution in BCN_11EE_. 

To further investigate the complex between BCN_n_ peptides and N-RNA, we made competition experiments between unlabeled peptides and fluorescent peptides of same length and/or charge, using the P_10EE_/BCN_10EE_ and pP_11_/BCN_11EE_ pairs. The unlabeled peptides were able to fully displace their fluorescent equivalents (Figure 8C), and the inhibition curves could be fitted with a Hill equation (Figure 8D). The K_i_s of unlabeled peptides were in the µM range (Table 5).

To probe the potency of M76 as an inhibitor, we carried out competition experiments with the M76 molecule and several BCN_n_ peptides. M76 was able to displace the fluorescent probes from N-RNA, and inhibition curves were obtained (Figure 8E). Since we used different N concentrations to have similar dynamic ranges of FP for the series of fluorescent peptides, we cannot directly compare M76 IC_50_s. However, K_i_s may be compared. K_i_ values were of the same order of magnitude, 0.5–1.8 µM, for peptides longer than B_6E_ (Table 5), suggesting that the inhibition mechanism by M76 is the same for these BCN_n_ peptides. Inhibition of BCN_5E_ yielded slightly higher K_i_ values (3.6–4.8 µM), but the K_d_ determined from a non-saturating binding curve might bias the result. M76 appeared to be more potent than the P_10EE_ peptide to displace BCN_10EE_ from N-RNA with a K_i_ of 1.1 versus 2.9 µM (Table 5). 

## 3. Discussion

Recognition of the RSV nucleocapsid by the RNA polymerase proceeds via a specific interaction between the essential RSV phosphoprotein polymerase cofactor P and the RSV nucleoprotein N in complex with genomic RNA. This interaction relies on the most C-terminal residue of P, Phe_241_, which inserts into a pocket at the surface of N. This view is supported by X-ray crystal structures of phenylalanine or short C-terminal P peptides in complex with the RSV N_NTD_ construct, revealing the Phe_241_ aromatic moiety as a main structural element that drives P binding thanks to well-defined interactions with N atoms delineating a binding pocket. Recently, Phe_241_ was further confirmed to be the main determinant for P anchoring to N by in silico energetic analysis and mutational analysis of Phe_241_ in vitro [59]. This view is also supported by in vitro binding and polymerase activity assays. While the aromatic Phe_241_Trp substitution maintained N binding, deletion of Phe_241_ and substitution of Phe_241_ by smaller amino acids like alanine and aspartate impaired N binding [31]. In vitro polymerase activity, as assessed with a minigenome, was completely abrogated by the Phe_241_Ala mutation [21]. However, even though Phe_241_ acts as a linchpin, adjacent residues also significantly contribute to the RSV N-P complex. The Leu_238_Ala and the double Glu_239_Ala/Asp_240_Ala mutants reduced minigenome activity 2-fold [21]. Early investigations led to the conclusion that a tract of nine C-terminal residues was necessary and sufficient for N binding [31]. Later we showed that a 12-mer peptide displayed approximately the same affinity in vitro as the full P_CTD_ domain, with K_d_ of 30–55 µM in salty buffer [33]. Affinity is expected to be rather weak, due to the requirement for polymerase processivity to elongate the newly synthesized RNA. Structural analysis of the RSV phosphoprotein by solution NMR indicated that P_Ctail_ remained unstructured in solution in the context of full-length P [26]. In contrast to other linear protein-binding motifs in P, like the binding sites for N^0^, M2-1 or L proteins, that transiently fold into α-helices and are stabilized in complex, P_Ctail_ did not display any propensity for any secondary structure, neither α-helical nor extended [29]. This is a strong indication that P_Ctail_ does not adopt any secondary structure in complex with N protein either. This implies that the entropic cost of binding likely remains moderate and that nearly all C-terminal amino acids of P may individually contribute to strengthen the N-P complex, even if they do not adopt a defined position.

The amino acid sequence of RSV N protein is rather conserved, and thus of particular interest for drug design [60,61]. The RSV N protein had already been identified before as a target for post-entry inhibitors. Notably the benzodiazepine RSV604, which reached phase II clinical trial before being discontinued, was proposed to bind to the N protein, since it elicited escape mutants on N [62,63]. Recently, EDP-938, another benzodiazepine in phase 2a clinical trial, was reported to target RSV N, also eliciting several mutations on N close to the P-binding pocket [42] leading to resistance. Based on the inhibitory potential of the M76 molecule we have suggested that the RSV P-binding pocket on N might be druggable [33]. This hypothesis was based on the assumption that a suitable inhibitor would be a direct competitor of the Asp_240_Phe_241_ dipeptide. If the interaction surface of the N-P complex extends beyond the Asp_240_Phe_241_ binding site and if up to 10 residues away from Phe_241_ contribute to the interaction energy, this raises the question if targeting the P-binding site with a small molecule would be sufficient to inhibit the N-P complex and displace P_Ctail_ and full-length P. In turn, this raises the question about how to design a meaningful assay to screen for inhibitors of the RSV N-P interaction. To address this question we sought for a more comprehensive view of the fuzzy complex formed between RSV N and C-terminal peptides to propose a relevant model of this interaction.

An inhibition assay by fluorescence polarization relies on the size difference between a free fluorescent probe and its complex with a target protein. We tested two forms of the nucleoprotein, the N_NTD_ domain and the N-RNA complex. Our results suggest that the N_NTD_ domain, although suitable in terms of size, induces binding biases, in particular one or more secondary interaction sites and binding of fluorescent labels to the P-binding pocket. In the N-RNA complex, which is structurally close to the nucleocapsid, the N protein surface is not fully accessible, which reduces unspecific binding, and even the P-binding site appears to be rather occluded (Figure 1B,C). N-RNA thus appears to be more relevant than N_NTD_ and the monomeric RNA-free N^0^-like N mutant used by Shapiro et al. [45]. Interestingly, Shapiro et al. had suggested a difference in binding affinity between BODIPY FL-labeled and unlabeled P_11_ by comparing FP and SPR data, which hints at binding of the fluorescent label [45]. Here, we show that binding of BODIPY FL remains moderate as compared to fluorescein, and that the peptide part of BCN_n_ peptides longer than 5-mers provides the main driving force for N-RNA binding. Steric hindrance in the N-RNA complex context could thus act as a filter to discriminate between specific and unspecific N binding. 

Since C-terminal RSV P-peptides are highly negatively charged, electrostatic interactions likely play an important role for the affinity of the RSV N-P complex. We confirmed the role of electrostatic interactions by comparing binding affinities for several fluorescent P peptides in salty versus salt-free buffers. In salt-free buffer, sub-micromolar affinity could be achieved. This is an asset for an FP assay to eliminate low binding molecules and rank high-affinity inhibitors in an HTS inhibition assay. Shapiro et al. reported a 20-fold increase in affinity by phosphorylation of Ser_232_ and Ser_237_ residues in the P_11_ peptide [45]. We observed similar effect on affinity by using phosphomimetic peptides, which exacerbate electrostatic interactions with N.

We previously assessed that the N-P complex was fuzzy, with the C-terminal Phe_241_ serving as a main anchor. To illustrate the dynamics at the P-binding site, we built a 3D structural model of the N_NTD_-P_11_ complex by docking the P_11_ peptide onto N_NTD_, using the Haddock webserver (Figure 9A–C). For this purpose, we used a structure of N_NTD_ extracted from the N_NTD_-P_2_ complex X-ray structure (PDB 4uc9) and generated structural models of P_11_ using the PEP-FOLD3 server [64]. The structural ensemble of free P_11_ was highly disordered, with the exception of the C-terminus that displays an incomplete α-helical turn involving residues Glu_239_-Phe_241_ (Figure 9D). The 9 N-terminal residues of P_11_ (Asp_231_-Glu_239_) were allowed to remain flexible during docking. Docked P_11_ also displayed a high degree of disorder, and the α-helical turn partly unwound (Figure 9C). Overall, docked P_11_ structural models were more extended than free P_11_ models, but it cannot be excluded that this is due to the difference of algorithms implemented in Haddock and PEP-FOLD3.

The electrostatic surface potential of N_NTD_ does not reveal a unique positively charged surface area, but rather a number of anchoring points afforded by positively charged residues. P_11_ displays neither defined conformation nor position, but seems to adopt a preferential orientation towards a positively charged patch located near the exit the cavity occupied by Phe_241_ (Figure 9A). This structural model provides a rationale for the overall increase of binding affinity observed with increasing peptide length up to 11 amino acids. Each amino acid contributes to binding affinity, either by electrostatic or van der Waals interactions. It also explains the concomitant decrease of FP in fluorescently labeled P peptides. As the label is attached to the opposite N-terminal side of the peptide, the fluorescent label experiences increased motional freedom, when the length of the tether increases. As the majority of C-terminal P peptide residues is charged, it is not surprising that a decrease in FP is observed in buffers with higher ionic strength.

As compared to difluoro BODIPY FL-labeled peptides, dicyano BODIPY FL-labeled peptides were more stable. The FP dynamic range was more extended for BCN_n_ peptides than for F_n_ peptides: This is an advantage for an FP assay. An FP assay relies on a compromise between peptide length and mobility of the fluorescent label. In this respect, 10-mer or 11-mer BCN_n_ peptides, which bind with sub-micromolar affinity in salt-free buffer and offer an extended dynamic FP range, appear to be suitable. Finally, M76 proved to be a suitable benchmark to screen for inhibitors, as it is able to fully displace even longer fluorescent P peptides.

## 4. Materials and Methods

### 4.1. Materials

All commercially available reagents and solvents were used as received unless otherwise noted. Fluorescein and fluorescein sodium salt were purchased from Sigma-Aldrich. Fluorescein-5-isothiocyanate (FITC) and all other reagent used for peptide synthesis were purchased from Fluorochem (Hadfield, UK) and Iris Biotech GmbH (Marktredwitz, Germany). pP_11_ and P_11SS_ peptides (>95% purity assessed by HPLC) were purchased from GeneCust (Luxemburg). 1-(2,4-dichlorobenzyl)-1H-pyrazole-3,5-dicarboxylic acid (M76) was purchased from ChemBridge. 2-Chlorotritylchloride resin (theoretical loading of 1.6 mmol/g) was obtained from Merck (Molsheim, France). Flash chromatography purifications were performed using the automated chromatography Reveleris^®^ Flash System (Grace, Büchi, Villebon-sur-Yvette, France) using prepacked normal phase cartridges from Interchim. Purifications were tracked with a dual λ absorbance UV detector and an ELSD detector.

HPLC analyses were performed on Hypersil C18 column (120 Å, 5 μm, 150 × 4.6 mm) using Waters Alliance 2690 separation module equipped with a Single Quadrupole Detector 2 (ESI quadrupole mass spectrometer), an ELS detector (Waters 2420) and a photodiode array detector (Waters 996). Reversed-phase ultra performance liquid chromatography-mass spectrometry (RP-UPLC-MS) analyses were performed on Waters equipment consisting of an ACQUITY UPLC H-Class separation module, photodiode array detector (eLambda detector), and a ESI triple quadrupole mass spectrometer (TQ detector). The analytical column used was the ACQUITY UPLC BEH C18 column (130 Å, 1.7 μm, 2.1 mm × 50 mm) operating at 0.6 mL·min^−1^ with linear gradient programs in 2.5 min run time (classical program: 5 to 100% of B in 2.5 min). UV monitoring was performed most of the time between 200 and 500 nm and was extracted at 214 nm. Solvent A consisted of H_2_O containing 0.1% (*v*/*v*) formic acid and solvent B was CH_3_CN containing 0.1% (*v*/*v*) FA. RP-HPLC purifications were performed on Waters equipment consisting of a 2545 quaternary pump, a photodiode array detector (Waters 2998), and an injector collector (Waters 2767). The preparative column, XBridge BEH C18 column (130 Å, 5 μm, 30 mm × 150 mm) was operated at 30 mL·min^−1^ with linear gradient programs in 15 min run time. The semi-preparative column, XBridge BEH C18 column (130 Å, 5 μm, 10 mm × 150 mm) was operated at 5 mL·min^−1^ with linear gradient programs in 15 min run time. Solvent C consisted of H_2_O containing 0.1% (*v*/*v*) TFA and solvent D consisted of CH_3_CN containing 9.9% (*v*/*v*) H_2_O and 0.1% TFA. Classical focused programs on preparative or semi-preparative column: 10 or 20% (*v*/*v*) slope of D in 15 min. Water was of Milli-Q quality and was obtained after filtration of distilled water through a Milli-Q cartridge system. CH_3_CN, FA, and TFA were of spectroscopic grade. High resolution mass spectra (HRMS-ESI) were obtained on a LCT Waters XE mass spectrometer equipped with an electrospray ionisation source. NMR spectra were performed on Bruker Avance spectrometers operating at 699 MHz for ^1^H NMR, 176 MHz for ^13^C NMR, 282 MHz for ^19^F and 160 MHz for ^11^B NMR experiments. The chemical shifts are reported in ppm relative to tetramethylsilane with the solvent resonance as the internal standard. Multiplicities were given as: s (singlet); d (doublets); t (triplets); q (quadruplets) m (multiplets). Coupling constants are reported as a J value in Hz.

### 4.2. Synthesis of Fluorescent Molecules

The fluorescein methyl ester **2** and ethyl ester **3** were synthetized in 73% and 40% yield from fluorescein according to Lu et al. [67] and C. Y. Ng et al. [68], respectively. Difluoro BODIPY FL **4** was synthesized according to Gießler et al. [69]. The dicyano BODIPY FL **5** was synthesized in two steps from the difluoro BODIPY FL methyl ester (Figure 1). The latter was obtained according to K. Gießler et al. [69].

Synthesis of methyl 3-(5,5-dicyano-7,9-dimethyl-5H-4λ^4^,5λ^4^-dipyrrolo [1,2-*c*:2′,1′-*f*][1,3,2]diazaborinin-3-yl)propanoate (**7**): BF_3_·OEt_2_ (34.0 µL, 0.27 mmol) was added to a cooled solution of difluoro BODIPY FL derivative **6** (415 mg, 1.36 mmol) in anhydrous CH_2_Cl_2_ (70 mL) at 0 °C. The mixture was then stirred at 25 °C for 10 min and TMSCN (851 µL, 6.80 mmol) was added. The reaction mixture was stirred at room temperature for 2 h. A saturated aqueous NaHCO_3_ solution was added. The organic phase was washed with water and was dried over Na_2_SO_4_. The solvent was evaporated under reduced pressure to give the titled product as a red solid (0.43 g, 98%). ^1^H NMR (699 MHz, CDCl_3_) δ 7.22 (s, 1H), 7.06 (d, *J* = 4.30 Hz, 1H), 6.43 (d, *J* = 4.30 Hz, 1H), 6.31 (s, 1H), 3.73 (s, 3H), 3.45 (t, *J* = 7.20 Hz, 2H), 2.88 (t, *J* = 7.30 Hz, 2H), 2.74 (s, 3H), 2.31 (s, 3H); ^13^C NMR (176 MHz, CDCl_3_) δ 172.5, 161.2, 158.1, 145.3, 133.6, 131.7, 129.7, 126.02 (q, *J*_CB_ = 75.2 Hz, 2 CN), 124.9, 122.1, 117.8, 52.3, 32.3, 24.3, 16.01, 11.7; ^11^B NMR (160 MHz, CD_2_Cl_2_) δ −16.86 (s); ^19^F NMR (282 MHz, CDCl_3_) no peaks observed. HRMS [ESI]: *m*/*z* calculated for C_17_H_17_BN_4_O_2_ [M − H]^−^ 319.1372; found: 319.1372.

Synthesis of 3-(5,5-dicyano-7,9-dimethyl-5H-4λ^4^,5λ^4^-dipyrrolo[1,2-c:2′,1′-f][1,3,2]diazaborinin-3-yl)propanoic acid (**5**): Concentrated HCl (16 mL) was added to a solution of methyl ester **7** (0.20 g, 0.62 mmol) in a mixture of THF (40 mL) and water (28 mL). The solution was stirred for 12 h at room temperature. Water (50 mL) and CH_2_CL_2_ (50 mL) were then added. The aqueous phase was extracted with CH_2_Cl_2_ (3 × 50 mL) and the combined organic phases were washed with brine and were dried over Na_2_SO_4_. The solvent was evaporated under reduced pressure to give the titled product as a red solid (0.17 g, 90%). M.p.: 211–232 °C. ^1^H NMR (500 MHz, DMSO-*d*_6_) δ 12.43 (s, 1H), 7.99 (s, 1H), 7.34 (d, *J* = 4.30 Hz, 1H), 6.63 (d, *J* = 4.20 Hz, 1H), 6.58 (s, 1H), 3.20 (t, *J* = 7.50 Hz, 2H), 2.82 (t, *J* = 7.50 Hz, 2H), 2.64 (s, 3H), 2.33 (s, 3H). ^13^C NMR (176 MHz, DMSO-*d*_6_) δ 173.1, 159.7, 157.5, 145.8, 132.8, 131.3, 130.3, 126.7, 125.5 (q, *J*_CB_ = 74.0 Hz, 2CN), 121.9, 117.7, 31.2, 23.7, 15.1, 11.1. ^11^B NMR (160 MHz, CD_2_Cl_2_) δ −17.0 (s). ^19^F NMR (282 MHz, DMSO-*d*_6_) no peaks observed. HRMS [ESI]: *m*/*z* calculated for C_16_H_15_BN_4_O_2_ [M − H]^−^ 305,1288; found: 305.1198.

NMR spectra are given in Appendix A. HPLC analysis of **5** and **7** is shown in Appendix A.

### 4.3. Peptide Synthesis

The BODIPY FL-labeled P peptide BF_2_pP11 was synthesized from purchased pP_11_ peptide and BODIPY FL as described in [45].

P_3_, P_4_, P_5E_, P_5S_, P_7S_, P_7E_, P_9E_, P_9S_, P_11SE_ and P_11EE_ peptides and their fluorescent dicyano BODIPY FL conjugates BCN_5E_, BCN_6E_, BCN_7E_, BCN_8E_, BCN_9E_, BCN_10EE_, BCN_11SE_, BCN_11EE_ were synthesized as described below.

Assemblies of protected peptides were carried out manually on 2-Chlorotritylchloride resin (theoretical loading of 1.6 mmol/g—0.2 g) using the Fmoc/*t*Bu strategy in a polypropylene reaction vessel fitted with polyethylene frits. After swelling of the resin for 15 min in CH_2_Cl_2_, the first amino acid was loaded through nucleophilic substitution by shaking the resin in a solution of Fmoc-Phe-OH (1 mmol/g) and DIPEA (1.5 equiv) in anhydrous CH_2_Cl_2_ (10 mL/g resin) for 30 min at room temperature. The un-reacted sites were then capped by shaking the resin in a mixture of MeOH/DIPEA/CH_2_Cl_2_ (2/1/17 *v*/*v*/*v*; 10.0 mL/g resin) for 10 min (repeated two times). The resin was then washed two times with DMF and three times with CH_2_Cl_2_ (10.0 mL/g resin) for 30 s. The overall process yielded a resin loading of around 0.65 mmol/g [determined from the dosage of the Fmoc released under the conditions of procedure A]. The number of reagent equivalents used in the following procedures is calculated from this resin loading. This step is followed by Fmoc removal according to procedure A. The elongation of the peptide was performed by cycle repetition of peptide coupling and Fmoc removal according to procedures B and A, respectively. Coupling of dicyano BODIPY FL, if required, was performed on the resin at the end of the peptide elongation on the amino terminus of the peptide according to procedure C. Cleavage of the peptide from the resin was carried out according to procedure D.

#### 4.3.1. Procedure A: Fmoc Removal

N-α-Fmoc protecting groups were removed by shaking the resin in piperidine/DMF solution (1/4, *v*/*v*; 10.0 mL/g resin) for 5 min at room temperature. The process was repeated two times for 10 min, and the completeness of deprotection was monitored by ultraviolet absorption measurement of the filtrate at 299 nm. The resin was then washed five times with DMF and once with CH_2_Cl_2_ (10.0 mL/g resin) for 30 s.

#### 4.3.2. Procedure B: Coupling Steps

Coupling reactions were performed by shaking with a solution of *N*-α-Fmoc-protected amino acid (2 equiv), PyBOP (2 equiv) and DIPEA (4 equiv) in DMF (10.0 mL/g resin) for 30 min at room temperature. The resin was then washed three times with DMF and once with CH_2_Cl_2_ (10.0 mL/g resin) for 30 s. Completeness of the coupling was controlled by TNBS and KAISER tests.

#### 4.3.3. Procedure C: Coupling of Dicyano BODIPY FL

The resin was shaken with a solution of dicyano BODIPY FL **5** (1 equiv), DIPEA (4–8 equiv depending of the number of peptide acidic functions) and PyBOP (2 equiv) at room temperature in DMF (10.0 mL/g resin). The resin was then washed 14 times with DMF and 8 times with CH_2_Cl_2_ (10.0 mL/g resin) for 30 s.

#### 4.3.4. Procedure D: Resin Cleavage and Protecting Groups Removal

The resin was first washed five times with CH_2_Cl_2_ (10.0 mL/g resin) before shaking in a mixture of TFA/TIS/H_2_O MQ (95/2.5/2.5 *v*/*v*/*v*; 10.0 mL/g resin) for 2 h at room temperature and then filtrated. The resin was rinsed once with the same acidic mixture of TFA/TIS/H_2_O MQ (95/2.5/2.5 *v*/*v*/*v*) for 1 min. The recovered solutions were concentrated under reduced pressure, and white solid was obtained by precipitation, triturating, and washing three times with diethyl ether. These crude peptides and crude dicyano BODIPY FL peptides, obtained as TFA salts, were purified by RP-HPLC on preparative or on semi-preparative column, respectively, using focused gradients, and freeze dried before their analysis by RP-UPLC-MS.

### 4.4. Peptide Purification and Yields

#### 4.4.1. Unlabeled Peptides

UPLC traces and analysis for P_n_ peptides are given in Appendix A.

P_3_ peptide was obtained as a white powder (34.0 mg, 50%). Analytical UPLC tr = 0.87 min; ESI-MS (positive mode) calculated for [C_18_H_23_N_3_O_8_], 409.2; found *m*/*z*, 410.4 (M + H)^+^. [Focused gradient 5–25% of B in 15 min].P_4_ peptide was obtained as a white powder (49.0 mg, 59%). Analytical UPLC tr = 1.01 min; ESI-MS (positive mode) calculated for [C_24_H_34_N_4_O_9_], 522.2; found *m*/*z*, 523.5 (M + H)^+^. [Focused gradient 10–30% of B in 15 min].P_5E_ peptide was obtained as a white powder (59.0 mg, 59%). Analytical UPLC tr = 1.00 min; ESI-MS (positive mode) calculated for [C_29_H_41_N_5_O_12_], 651.3; found *m*/*z*, 652.6 (M + H)^+^. [Focused gradient 15–35% of B in 15 min].P_7E_ peptide was obtained as a white powder (43.0 mg, 33%). Analytical UPLC tr = 1.43 min; ESI-MS (positive mode) calculated for [C_39_H_57_N_7_O_16_], 879.4; found *m*/*z*, 880.8 (M + H)^+^. [Focused gradient 25–35% of B in 15 min].P_9E_ peptide was obtained as a white powder (27.0 mg, 17%). Analytical UPLC tr = 1.29 min; ESI-MS (positive mode) calculated for [C_47_H_68_N_10_O_21_], 1108.5; found *m*/*z*, 1109.7 (M + H)^+^. [Focused gradient 20–30% of B in 15 min].P_11EE_: SPPS of P_11EE_ peptide was performed on 150 mg of 2-CTC resin. The peptide P_11EE_ was obtained as a white powder (28.0 mg, 20%). Analytical UPLC tr = 1.28 min; ESI-MS (positive mode) calculated for [C_56_H_80_N_12_O_27_], 1352.5; found *m*/*z*, 1353.9 (M + H)^+^. [Focused gradient 20–30% of B in 15 min].P_11SE_: SPPS of P_11SE_ peptide was performed on 15.0 mg of 2-CTC resin. P_11SE_ was obtained as a white powder (4.50 mg) and was used without further purification for F_11SE_ synthesis.

#### 4.4.2. Dicyano BODIPY FL-Labeled BCN_n_ Peptides

UPLC traces and analysis for BCN_n_ peptides and blank are given in Appendix A.

BCN_5E_: Starting from P_5E_ on resin (8.71 µmol), BCN_5E_ was obtained as a red powder (2.63 mg, 32%). Analytical UPLC tr = 1.61 min; ESI-MS (negative mode) calculated for [C_45_H_54_BN_9_O_13_], 939.8; found *m*/*z*, 939.4 (M − H)^−^. [Focused gradient 38–48% of B in 15 min].BCN_6E_: Starting from P_6E_ on resin (8.71 µmol), BCN_6E_ was obtained as a red powder (1.60 mg, 18%). Analytical UPLC tr = 1.74 min; ESI-MS (negative mode) calculated for [C_51_H_65_BN_10_O_14_], 1052.5; found *m*/*z*, 1051.2 (M − H)^−^. [Focused gradient 43–53% of B in 15 min].BCN_7E_: Starting from P_7E_ on resin (8.71 µmol), BCN_7E_ was obtained as a red powder (2.22 mg, 22%). Analytical UPLC tr = 1.66 min; ESI-MS (negative mode) calculated for [C_55_H_70_BN_11_O_17_], 1167.5; found *m*/*z*, 1166.9 (M − H)^−^. [Focused gradient 40–50% of B in 15 min].BCN_8E_: Starting from P_8E_ on resin (8.71 µmol), BCN_8E_ was obtained as a red powder (2.00 mg, 18%). Analytical UPLC tr = 1.56 min; ESI-MS (positive mode) calculated for [C_59_H_76_BN_13_O_19_], 1282.6; found *m*/*z*, 1283.9 (M + H)^+^. [Focused gradient 36–46% of B in 15 min].BCN_9E_: Starting from P_9E_ on resin (8.71 µmol), B_9E_ was obtained as a red powder (1.17 mg, 10%). Analytical UPLC tr = 1.53 min; ESI-MS (positive mode) calculated for [C_63_H_81_BN_14_O_22_], 1396.6; found *m*/*z*, 1398.2 (M + H)^+^. [Focused gradient 35–45% of B in 15 min].BCN_10EE_: Starting from P_10EE_ on resin (8.71 µmol), BCN_10EE_ was obtained as a red powder (1.00 mg, 8%). Analytical UPLC tr = 1.50 min; ESI-MS (negative mode) calculated for [C_68_H_88_BN_15_O_25_], 1525.6; found *m*/*z*, 1525,1(M − H)^−^. [Focused gradient 34–44% of B in 15 min]. UV-Visible absorption spectrum and fluorescence excitation/emission spectra are given in Appendix A.BCN_11EE_: Starting from P_11EE_ on resin (8.71 µmol), BCN_11EE_ was obtained as a red powder (1.01 mg, 11%). Analytical UPLC tr = 1.49 min; ESI-MS (negative mode) calculated for [C_72_H_93_BN_16_O_28_], 1640.6; found *m*/*z*, 1640.3 (M − H)^−^. [Focused gradient 33–43% of B in 15 min].BCN_11SE_: Starting from P_11SE_ on resin (4.35 µmol), BCN_11SE_ was obtained as a red powder (1.12 mg, 16%). Analytical UPLC tr = 1.48 min; ESI-MS (negative mode) calculated for [C_70_H_91_BN_16_O_27_], 1598.6; found *m*/*z*, 1598.2 (M − H)^−^. [Focused gradient 32–42% of B in 15 min].

#### 4.4.3. Fluorescein-Labeled F_n_-Peptides

Due to reported instability of FTU-labelled peptides under acidic conditions [70], the coupling of FITC with synthesized P_3_, P_4_, P_5E_, P_7E_, P_9E_, P_11SE_, P_11EE_ peptides and commercial pP_11_ and P_11SS_ peptides was performed in solution. FITC (2 equiv) was added to a solution of the peptides (1 equiv) and DIPEA (9 equiv) in anhydrous DMF (0.06 mL/mg peptide). Solutions were stirred for 30 min to 1h depending on the peptide. Solutions were then purified by HPLC on semi-preparative column using focused gradients and freeze dried. UPLC traces and analysis for F_n_ peptides are given in Appendix A.

F_3_: Starting from P_3_ (5.00 mg, 9.55 µmol), F_3_ was obtained as a yellow powder (3.00 mg, 48%). Analytical UPLC tr = 1.31 min; ESI-MS (negative mode) calculated for [C_39_H_34_N_4_O_13_S], 798.2; found *m*/*z*, 797.6 (M − H)^−^. [Focused gradient 25–45% of B in 15 min].F_4_: Starting from P_4_ (5.00 mg, 7.85 µmol), F_4_ was obtained as a yellow powder (4.30 mg, 62%). Analytical UPLC tr = 2.12 min; ESI-MS (negative mode) calculated for [C_45_H_45_N_5_O_14_S], 911.3; found *m*/*z*, 910.6 (M − H)^−^. [Focused gradient 55–65% of solvent D consisted of MeOH containing 9.9% (*v*/*v*) H_2_O and 0.1% TFA in 15 min].F_5E_: Starting from P_5E_ (5.00 mg, 6.53 µmol), F_5E_ was obtained as a yellow powder (3.60 mg, 53%). Analytical UPLC tr = 0.98 min; ESI-MS (positive mode) calculated for [C_50_H_52_N_6_O_17_S], 1040.3; found *m*/*z*, 1041.8 (M + H)^+^. [Focused gradient 35–45% of B in 15 min].F_7E_: Starting from P_7E_ (5.00 mg, 5.03 µmol), F_7E_ was obtained as a yellow powder (2.60 mg, 50%). Analytical UPLC tr = 1.53 min; ESI-MS (positive mode) calculated for [C_60_H_68_N_8_O_21_S], 1268.4; found *m*/*z*, 1270.0 (M + H)^+^. [Focused gradient 30–50% of B in 15 min].F_9E_: Starting from P_9E_ (5.00 mg, 4.09 µmol), F_9E_ was obtained as a yellow powder (1.40 mg, 22%) Analytical UPLC tr = 1.42 min; ESI-MS (positive mode) calculated for [C_68_H_79_N_11_O_26_S], 1497.5; found *m*/*z*, 1499.7 (M + H)^+^. [Focused gradient 30–50% of B in 15 min].F_11_: Starting from commercial P_11_ (4.30 mg, 3.11 µmol), F_11SS_ was obtained as a yellow powder (3.5 mg, 68%). Analytical UPLC tr = 1.45 min; ESI-MS (positive mode) calculated for [C_73_H_87_N_13_O_30_S], 1657.5; found *m*/*z*, 1659.8 (M + H)^+^. [Focused gradient 25–45% of B in 15 min].F_11SE_: Starting from P_11SE_ (4.50 mg, 3.43 µmol), F_11SE_ was obtained as a yellow powder (0.90 mg, 15%). Analytical UPLC tr = 1.39 min; ESI-MS (negative mode) calculated for [C_75_H_89_N_13_O_31_S], 1699.6; found *m*/*z*, 1697.7 (M − H)^−^. [Focused gradient 25–45% of B in 15 min].F_11EE_: Starting from P_11EE_ (5 mg, 3.41 µmol), F_11EE_ was obtained as a yellow powder (2.40 mg, 40%). Analytical UPLC tr = 1.39 min; ESI-MS (positive mode) calculated for [C_77_H_91_N_13_O_32_S], 1741.6; found *m*/*z*, 1742.6 (M + H)^+^. [Focused gradient 25–45% of B in 15 min].pF_11_: Starting from commercial pP_11_ (4.20 mg, 2.93 µmol), pF_11_ was obtained as a yellow powder (1.10 mg, 20%). Analytical UPLC tr = 1.36 min; ESI-MS (negative mode) calculated for [C_73_H_89_N_13_O_36_P_2_S], 1817.5; found *m*/*z*, 908.7 (M − 2H)^2−^. [Focused gradient 25–45% of B in 15 min].

### 4.5. Bacterial Expression and Purification of RSV N Protein

Recombinant RSV N protein was produced in *E. coli* BL21(DE3) bacteria (Novagen, Madison, WI). The N-terminal domain N_NTD_ (residues 31–252), containing a C-terminal 6x histidine tag, was overexpressed using the pET-N(31–252) plasmid and purified as described previously [21]. In a final step N_NTD_ was dialyzed into MES 20 mM pH 6.5, NaCl 250 mM buffer [33]. Full-length N was produced by co-expression with the C-terminal domain of RSV P protein in *E. coli* BL21(DE3) co-transformed with pET-N and pGEX-P(161–241) plasmids [21]. Cultures were grown at 37 °C for 8 h in Luria-Bertani (LB) medium supplemented with 50 µg/mL kanamycin and 100 µg/mL ampicillin. An equivalent volume of LB was then added, and protein expression was induced with 80 µg/mL isopropyl-β-D-thio-galactoside (IPTG) for 15 h at 28 °C. Bacteria were harvested by centrifugation. The bacterial pellet was resuspended in lysis buffer (20 mM Tris-HCl pH 8.5, 150 mM NaCl, 1 mM EDTA, 2 mM dithiothreitol, 0.2% Triton X-100, 1 mg/mL lysozyme). Complete protease inhibitor cocktail (Roche, Mannheim, Germany) was added, and the suspension was incubated for 1 h on ice, sonicated and centrifuged at 4 °C for 30 min at 10,000× *g*. Glutathione-Sepharose 4B beads (GE Healthcare, Vélizy-Villacoublay, France) were added to the clarified supernatant and incubated at 4 °C for 3 h. The beads were washed twice in lysis buffer and three times in 20 mM Tris-HCl pH 8.5, 150 mM NaCl buffer. GST was cleaved from the N-P(161–241) complex by treating the beads with thrombin (Novagen) for 16 h at 20 °C. The supernatant was then loaded onto a Superdex 200 16/30 column (GE Healthcare) and eluted in 20 mM Tris-HCl pH 8.5, 150 mM NaCl buffer. Finally, the fractions containing the N protein in the form of an N-RNA complex were pooled and concentrated up to 2 mg/mL (45 µM of N protein). The protein was subsequently dialyzed into 20 mM Tris pH 8.0 buffer containing 0.01% Brij-35. The concentration of N protein was determined with a BCA assay (Thermo Fisher, Illkirch-Graffenstaden, France), calibrated with BSA.

### 4.6. NMR Measurements

NMR titration experiments with ^15^N-N_NTD_ were performed in 20 mM MES pH 6.5, 250 mM NaCl buffer. pP_11_ peptide was solubilized at 1 mg/mL in MQ water by addition of 1 M NaOH until the pH became neutral. pP_11_ aliquots were lyophilized. For the titration experiment, pP_11_ was added stepwise to a 50 µM ^15^N-N_NTD_ solution, using 0.2–6.2 molar equivalents. NMR data were acquired on a 14.1 T (600 MHz ^1^H frequency) Bruker Avance III NMR spectrometer equipped with a cryogenic TCI probe. A standard 2D ^1^H-^15^N HSQC spectrum was recorded for each titration point.

For titration with fluorescein, 0.5–8 molar equivalents of fluorescein were added from a 10 mM stock solution in water at pH 6.5 to 55 µM ^15^N-N_NTD_. NMR measurements were performed on a 16.4 T (700 MHz ^1^H frequency) Bruker NMR spectrometer equipped with a NEO console and a cryoTXO probe. All samples contained 7.5% ^2^H_2_O to lock the spectrometer frequency and 100 µM DSS to reference ^1^H chemical shifts. The temperature was set to 20 °C. NMR data were processed with TopSpin 4.0 (Bruker Biospin, Wissembourg, France) and analyzed with CcpNmr Analysis 2.4 software [71]. Backbone chemical shift assignment of N_NTD_ was done previously [33].

Combined amide ^1^H and ^15^N chemical shift perturbations Δδ_HN_ were calculated with a scaling factor of 1/10 for ^15^N, which corresponds to the ratio of ^15^N and ^1^H gyromagnetic ratios (Equation (1)):(1)$${\Delta \delta }_{\mathrm{HN}}=\sqrt{\left({\left(\delta^{1}H-\delta^{1}H_{\mathrm{ref}}\right)}^{2}+{\left(\delta^{15}N-\delta^{15}N_{\mathrm{ref}}\right)}^{2}/100\right)}$$

^1^H and/or ^15^N chemical shift perturbations were fitted in CcpNmr Analysis 2.4 as a function of the molar ligand:N_NTD_ ratio r, using a single binding site model with a 1:1 stoichiometry, and assuming a fast chemical exchange regime (Equation (2)). K_d_ is the dissociation constant of the complex with N_NTD_. δ_free_ and δ_bound_ (in ppm) are the chemical shifts of free and bound N_NTD_.
(2)$$\left(\delta -\delta_{\mathrm{ref}}\right)=\frac{1}{2}\left(\delta_{\mathrm{bound}}-\delta_{\mathrm{free}}\right)\times \left(\frac{K_{d}}{{\left[N_{\mathrm{NTD}}\right]}_{\mathrm{tot}}}+1+r-\sqrt{{\left(\frac{K_{d}}{{\left[N_{\mathrm{NTD}}\right]}_{\mathrm{tot}}}+1+r\right)}^{2}-4r}\right)$$

For residues in intermediate exchange regime, the exchange rate between free and bound protein states k_ex_ (s^−1^) was estimated according to Equation (3), where BF is the Larmor frequency (MHz) of the observed nucleus.
(3)$$k_{ex}=\pi \times \Delta \nu =\pi \times \left(\delta_{\mathrm{bound}}-\delta_{\mathrm{free}}\right)\times \mathrm{BF}\;$$

### 4.7. Fluorescence Polarization Measurements

Fluorescence polarization measurements were carried out on a Paradigm Detection Platform (Beckman Coulter, Brea, CA, USA) operating with a microplate reader, a fluorescein detection cartridge (excitation range 485/20 nm, emission range 535/25 nm for both parallel and perpendicular components) and the SpectraMax Pro 6.1 software (Molecular Devices, San Jose, CA, USA). Samples were placed in polystyrene black flat bottom 96-well microplates (Cellstar, Greiner bio-one, Frickenhausen, Germany) with 20 µL final volume in each well. If not stated otherwise, the final concentration of fluorescent ligands was 200 nM. The buffer was either 20 mM MES pH 6.5, salt-free or with 250 mM NaCl, or 20 mM Tris pH 8.0 salt-free or with 100 mM NaCl, supplemented with 0.01% Brij-35. Mixing was done using the corresponding plate reader option. The temperature was set to 20 °C. Data were recorded in top reading mode. Measurements were done in triplicate and after an incubation time of 30 min, if not indicated otherwise. Fluorescence polarization (FP) in mFP units was determined from the parallel (I_∥_) and perpendicular (I_⊥_) components of the emitted light according to Equation (4) [43]. A grating factor G of 1.2 was determined using 1 nM fluorescein in 1 mM NaOH and assuming a theoretical FP of 27 mFP [48]. Background signal was measured from wells containing only buffer, and the mean value was subtracted from each I_∥_ and I_⊥_ component.
(4)$$\mathrm{FP}\;\left(\mathrm{mP}\right)=1000\times \left(I_{∥}-{G*I}_{⊥}\right)/\left(I_{∥}+G\times I_{⊥}\right)$$

Theoretical fluorescence polarization values of a fluorophore bound to N protein were calculated from the fluorescence lifetime τ and the molecular weight MW using Perrin’s equation (Equation (5)). A is the fluorescence anisotropy related to FP by Equation (6) [43], A_0_ the intrinsic anisotropy (0.4), T the temperature in Kelvin, k the Planck constant, η the dynamic viscosity, and V_h_ the hydrated volume. V_h_ was calculated according to Equation (7), where N is the Avogadro number, v_2_ the partial specific volume of the protein (7.34 × 10^−7^ m^3^/g), δ the degree of hydration (0.75), and v_1_^0^ the specific volume of water (10^−6^ m^3^/g). We used A_0_, v_2_, δ and v_1_^0^ values reported for bovine serum albumine [72].
(5)$$A=A_{0}/\left(1+\frac{\mathrm{kT}\times \tau }{{\eta *V}_{h}}\right)$$
(6)FP=1000×3A/(2+A)
(7)$$V_{h}=\left(\mathrm{MW}/N\right)\times \left(v_{2}+\delta \times v_{1}^{0}\right)$$

We assumed that the measured fluorescence polarization FP is a linear combination of the FP of the fluorescent probe in the absence of N protein (FP_min_) and the FP of N-bound fluorescent probe (FP_max_), weighted by the molar fraction. The molar fraction x = [P]_bound_/[P]_0_ can then be expressed as a ratio of FP differences (Equation (8)).
(8)$$x=\left(\mathrm{FP}-{\mathrm{FP}}_{\min}\right)/\left({\mathrm{FP}}_{\max}-{\mathrm{FP}}_{\min}\right)=∆\mathrm{FP}/∆{\mathrm{FP}}_{\max}\;$$

To determine the K_d_ value of a complex between a fluorescent probe and N protein, a series of samples with constant probe concentration and varying N concentration was measured. FP binding curves were fitted with a single binding site model and a 1:1 stoichiometry. The molar fraction of bound fluorescent probe was expressed as a function of the ratio of protein versus probe concentrations, r = [N]_0_/[P]_0_, according to Equation (9). Binding curves were fitted in Origin 7 software according to Equation (10), obtained by combining Equations (8) and (9).
(9)$$x=0.5\times \left[1+r+{\;K}_{d}/{\left[P\right]}_{0}-\sqrt{{\left(1+r+K_{d}/{\left[P\right]}_{0}\right)}^{2}-4r}\right]$$
(10)$$∆\mathrm{FP}=0.5\times ∆{\mathrm{FP}}_{\max}\times \left[1+r+{\;K}_{d}/{\left[P\right]}_{0}+\sqrt{{\left(1+r+K_{d}/{\left[P\right]}_{0}\right)}^{2}-4r}\right]$$

To assess inhibition of a complex between a fluorescent probe and N protein by peptides or the M76 molecule, the concentration of N was set to work with an FP range comprised between FP_min_ and ~80% FP_max_. Peptides were added from 1 or 2 mM stock solution in water at neutral pH. M76 was added from 10 or 20 mM stock solutions in ethanol or in DMSO. The percentage of inhibition (%_i_) was calculated as the relative change in FP (Equation (11)). %_i_ was fitted as a function of inhibitor concentration with a Hill equation, where IC_50_ is the inhibitor concentration resulting in 50% inhibition (Equation (12)). Inhibition data were fitted with Origin 7, and in parallel with the Excel solver module. All data were compatible with an absence of cooperativity. FP_max_ and FP_min_ were fitted, and the maximal inhibition %_i,max_ was set to 100%.
(11)$${\%}_{i}=100\times \left[1-\left(\mathrm{FP}-{\mathrm{FP}}_{\min}\right)/\left({\mathrm{FP}}_{\max}-{\mathrm{FP}}_{\min}\right)\right]$$
(12)$${\%}_{i}={\%}_{i,\max}\times \frac{\left[\mathrm{inhibitor}\right]}{\left[\mathrm{inhibitor}\right]+{\mathrm{IC}}_{50}}$$

Inhibition constants K_i_ were calculated on the IC50-to-Ki converter for a protein-ligand-inhibitor system (https://bioinfo-abcc.ncifcrf.gov/IC50_Ki_Converter/index.php, accessed on 24 August 2022) [49], assuming a competitive mechanism (Equation (13)). This tool takes into account the free concentrations of protein and ligand.
(13)$$K_{i}=\left({\mathrm{IC}}_{50}-\frac{\left[\mathrm{protein}\right]}{2}\right)/\left(\frac{\left[\mathrm{peptide}\right]}{K_{d}}+1\right)$$

A Z’-factor for FP measurements without test compound [58] was calculated from 7 data points for the condition 200 nM B_10EE_ and 1 µM N-RNA, using Equation (14), where σ_+_ and σ_-_ are the standard deviations of positive (fluorescent peptide bound to N protein) and negative (free fluorescent peptide) controls, respectively, and μ_+_ and μ_-_ their mean values.
(14)$$Z^{\prime }=1-\left(3\sigma_{+}+3\sigma_{-}\right)/\left|\mu_{+}-\mu_{-}\right|$$

### 4.8. Complex Modelling with Haddock

Docking of the P_11_ peptide onto N_NTD_ was performed with the Haddock 2.4 software [65] on the WeNMR server [73]. A structure of P_11_ was built with the PEP-FOLD3 server [64] and allowed to be flexible (residues 1–9) during docking. The C-terminal Asp_10_ and Phe_11_ residues (equivalent to Asp_240_ and Phe_241_) were declared as active residues. The structure of N_NTD_ was extracted from the X-ray structure of the N_NTD_-P_2_ complex (PDB 4uc9). N residues 46, 50, 53, 128, 131, 132, 135, 145, and 151, lining the P-binding pocket, were declared as active residues. Passive residues were automatically defined around active residues. Default scoring parameters for protein–protein complexes were used, except for Evdw 1, which was increased from 0.01 to 1. 1000 initial structures were generated. 200 final structures were refined in water and clustered according to the RMSD criterion. Haddock clustered 156 structures in 12 clusters. Cluster 1 with the best haddock score (−96.4 ± 4.9) contained 63 structures, i.e., 40% of clustered structures.

### 4.9. Docking of Small Compounds

Fluorescein and fluorescein methyl ester were docked on the P-binding pocket of N_NTD_ (PDB 4ucc) using Smina software and MOE [53]. For Smina, Vinardo [54] scoring function was selected with an exhaustiveness of 8. The pose with the best scoring value was retained. For MOE, the placement method used was Triangle Matcher with London dG score and the number of poses was set to 30. These poses were then refined with GBVI/WSA dG score with rigid receptor, and the number of output poses was set to 5. The pose with the best GBVI/WSA dG score was retained.

## Data Availability

Not applicable.

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
