# Peer review of "Investigation of the Fuzzy Complex between RSV Nucleoprotein and Phosphoprotein to Optimize an Inhibition Assay by Fluorescence Polarization"

_ijms, 2022, doi:10.3390/ijms24010569_

Round 1

Reviewer 1 Report

The manuscript presented by Khodjoyan and coauthors describes a very interesting study of the investigation of the fuzzy complex between RSV nucleoprotein and phosphoprotein to optimize an inhibition assay by fluorescence polarization. The work shows a great quality of presentation and scientific soundness; therefore, I consider the manuscript accepted for publication.

There is one point that is worth to comment. If the peptide structure was built on the PEP-FOLD3 server, which structure did the peptide show at the start of the docking calculations? And if an alpha helix structure was predicted, did it unfold during the docking calculations?

Just a minor revision, it is mentioned in the text "...which ranged from 0.1 to 120 μM (Figure 3A,D).", but there is no plot D in Figure 3.

Author Response

We thank reviewer 1 for positive comments and for valuable suggestions to improve the manuscript.

We removed the mention to Fig3 D, which does not exist.

Following the comment of reviewer 1, we have added a short discussion about the structural ensemble of the P11 peptide, obtained with PEP-FOLD3 and used for Haddock docking. P11 models were mainly disordered. Interestingly, structural alignment of the C-termini (3 last residues, Glu239-Phe241) shows an incomplete C-terminal helix turn. The 9 N-terminal residues of P11 were allowed to be flexible during the docking procedure, which resulted in unwinding of this helix turn in several models. To illustrate our point, we have added two panels to Figure 9, showing the structural ensemble generated with PEP-FOLD3 (panel D) and the comparison between the starting model used for P11 and the ensemble of docked models (panel C). The panels in Figure 9 have been slightly rearranged, and the legend was changed accordingly.

Reviewer 2 Report

Respiratory syncytial virus (RSV) is an important human pathogen and there is a constant effort to better understand and fight RSV infection.

The authors optimized an assay to perform high-throughput screens (HTS) to identify molecules inhibiting the essential interaction between the viral nucleoprotein (N) and the C-terminal domain of the phosphoprotein (PCTD). They further characterized this interaction and their data reinforces the idea that other residues than the last two amino acids of P also play a role in the binding. The authors tested a wide range of conditions and propose optimal conditions to screen antiviral molecules.

This work is of interest both to researchers studying RSV and to laboratories developing similar HTS.

The manuscript is well-written and easy to follow.

Minor comments:

- In Figure 2C, it could be nice to add a panel with a surface representation (keeping the same color code) to see the binding pocket of PCTD.

- Check the spelling of PCTD vs PCDT

- Line 236, Figure 3D does not exist. 

Author Response

We thank reviewer 2 for their positive comments and for valuable suggestions to improve the manuscript.

We removed the mention to Fig3 D, which does not exist.

We have also spell checked the manuscript (in particular for typos in PCTD).

Upon suggestion of reviewer 2, we have added a surface representation of the NNTD protein in Figure 2 and changed the legend accordingly.